# Chemical Markers and Pharmacological Characters of *Pelargonium graveolens* Essential Oil from Palestine

**DOI:** 10.3390/molecules27175721

**Published:** 2022-09-05

**Authors:** Nidal Jaradat, Mohammed Hawash, Mohammad Qadi, Murad Abualhasan, Aseel Odetallah, Ghfran Qasim, Reem Awayssa, Amna Akkawi, Ibtesam Abdullah, Nawaf Al-Maharik

**Affiliations:** 1Department of Pharmacy, Faculty of Medicine and Health Sciences, An-Najah National University, Nablus P.O. Box 7, Palestine; 2Department of Biomedical Sciences, Faculty of Medicine and Health Sciences, An-Najah National University, Nablus P.O. Box 7, Palestine; 3Department of Chemistry, Faculty of Sciences, An-Najah National University, Nablus P.O. Box 7, Palestine

**Keywords:** *Pelargonium graveolens*, phytochemical markers, antioxidant, key metabolic enzymes, antimicrobial, cytotoxic, cyclooxygenase

## Abstract

*Pelargonium graveolens* leaves are widely used in traditional medicine for relieving some cardiovascular, dental, gastrointestinal, and respiratory disorders. They are also used as food and tea additives in Palestine and many other countries. Consequently, this investigation aimed to describe the chemical markers, cytotoxic, antioxidant, antimicrobial, metabolic, and cyclooxygenase (COX) enzymes inhibitory characteristics of *P. graveolens* essential oil (PGEO) from Palestine utilizing reference methods. There were 70 chemicals found in the GCMS analysis, and oxygenated terpenoids were the most abundant group of the total PGEO. Citronellol (24.44%), citronellyl formate (15.63%), *γ*-eudesmol (7.60%), and iso-menthone (7.66%) were the dominant chemical markers. The EO displayed strong antioxidant activity (IC_50_ = 3.88 ± 0.45 µg/mL) and weak lipase and α-amylase suppressant effects. Notably, the PGEO displayed high α-glucosidase inhibitory efficacy compared with Acarbose, with IC_50_ doses of 52.44 ± 0.29 and 37.15 ± 0.33 µg/mL, respectively. PGEO remarkably repressed the growth of methicillin-resistant *Staphylococcus aureus* (MRSA), even more than Ampicillin and Ciprofloxacin, and strongly inhibited *Candida albicans* compared with Fluconazole. The highest cytotoxic effect of the PGEO was noticed against MCF-7, followed by Hep3B and HeLa cancer cells, with IC_50_ doses of 32.71 ± 1.25, 40.71 ± 1.89, and 315.19 ± 20.5 µg/mL, respectively, compared with doxorubicin. Moreover, the screened EO demonstrated selective inhibitory activity against COX-1 (IC_50_ = 14.03 µg/mL). Additionally, PGEO showed a weak suppressant effect on COX-2 (IC_50_ = 275.97 µg/mL). The current research can be considered the most comprehensive investigation of the chemical and pharmacological characterization of the PGEO. The results obtained in this study demonstrate, without doubt, that this plant represents a rich source of bioactive substances that can be further investigated and authenticated for their medicinal potential.

## 1. Introduction

Herbs have long been used in food and medicine due to their nutritional and therapeutic value. Even today, the modern pharmaceutical industry relies on plant extracts and the utilization of active compounds with a specific mode of action. An FDA evaluation of all novel molecular entities (NMEs) found that natural medicinal substances and their metabolites accounted for more than a third of all NMEs, with plants representing over a quarter of them. Plants have a wide range of biological activities on the human body, including antipyretic, sedative, anti-inflammatory, antioxidant, antimicrobial, and vasodilatory effects [1]. More studies into herbs and plant extracts can lead to breakthrough treatments for key health problems such as oxidative stress, diabetes, obesity, antibiotic resistance, and cancer. Many cardiovascular diseases (CVDs), including heart failure, atherosclerosis, and ventricular remodeling, are linked with producing reactive oxygen species (ROS) during excessive oxidative stress. A balance of pro-oxidants and antioxidants protects the genomic integrity of the cell. If this balance is disrupted, host immunity is compromised, which affects normal cellular signaling pathways and leads to uncontrolled cell proliferation. This can result in cancer and macrophage polarization, leading to atherogenic plaque formation [2].

Higher basal oxidative stress is observed under these conditions, particularly in the tumor environment, taking advantage of the upregulated antioxidant system. Antioxidants, including quercetin, Q10 coenzyme, β-carotene, and vitamins E and C, have preventive and curative properties in several types of CVD [3].

Obesity is a growing health issue that has nearly tripled globally since 1975 and was thought to be a problem only in rich countries but is now spreading rapidly to middle-and low-income areas [4]. In Palestine, the combined prevalence of obesity and overweightness is 6% in children and 18% in adults. Increased BMI is regarded as a key risk factor for many diseases, for instance, CVDs, diabetes type 2, and some types of cancers [5].

Diabetes mellitus and its complications are putting excessive strain on the healthcare system and patients and are regarded as major health issues. Diabetes affects an estimated about 400 million patients globally [6]. Diabetes was estimated to affect 9.7% of adults aged 25 and older in Palestine in 2000, rising to 15.3% by 2010, and is expected to reach 20.8% by 2020 and 23.4% by 2030 [7,8].

Antimicrobial resistance is now considered one of the most serious issues confronting the healthcare sector as a result of antimicrobial misuse and overuse; as a result, “superbugs” are spreading, with no antimicrobials available to treat them. The constant rise in antibiotic-resistant strains, as well as the harmful effects of conventional antimicrobials, have sparked a hunt for herbal alternatives [9].

Cancer is the second greatest cause of death worldwide, according to the World Health Organization (WHO), with 70% of deaths occurring in third-world nations [10]. Over the next few years, this percentage is likely to rise. Cancer is Palestine’s second-largest cause of death, accounting for 14% of all fatalities, second only to CVD. This high percentage is due, in part, to poor prevention and screening programs, late-stage diagnoses, and a lack of treatment facilities [11].

Non-steroidal anti-inflammatory agents (NSAIDs) are the most commonly prescribed drugs in the world, with their pharmacological effects derived from the inhibition of the cyclooxygenase (COX) enzyme, and they play an important role in the release of inflammatory mediators such as prostaglandins [12]. Usually, NSAIDs are utilized for the treatment of inflammation, pain, and fever caused by a variety of disorders. Non-selective NSAIDs enhance the risk of gastric ulcer, hemorrhage, CVD, gastrointestinal hemorrhage, and renal impairment. However, selective NSAIDs, such as COX-2 inhibitors, were developed to maintain analgesic efficacy, while the reductions in the adverse GI effects are associated with COX-1 inhibition [13].

*Pelargonium graveolens* L’Hér. is an aromatic herb in the Geraniaceae family that is native to a few geographical areas of South Africa, and mainly distributed throughout the Mediterranean region [14,15]. Around the world, PGEO is widely utilized as a pharmaceutical, cosmetic, and flavoring agent as well as in folkloric foods and aromatherapy industries [14,16,17]. It is one of the best oils utilized in skincare due to its effectiveness in cleaning oily skin and opening pores. The leaves are utilized in the form of herbal tea to cure tonsillitis, improve circulation, ease tension, fight anxiety, and de-stress [18]. Several studies have revealed that PGEO and extracts have antibacterial and antifungal properties, as well as high antioxidant activity, and can be utilized to heal a variety of illnesses including cancer, diabetes, and obesity [16]. Therefore, this study aims to explore the chemical contents, antiobesity and antidiabetic, cytotoxicity, and COX inhibitory properties of PGEO grown in Palestine, which have never before been examined in this way.

## 2. Results

### 2.1. Phytochemistry

The hydrodistillation of *P. graveolens* dried leaves yielded a colorless oil with a yield of 1.01%. GC-MS was used to investigate the chemical composition of the EO, allowing the identification and quantification of 70 components that accounted for 99.47% of the total EO Table 1 (Figure 1). Table 1 shows the names, retention times, retention indices (RI), and percentages of identified compounds obtained by GC-MS for EOs.

The retention time of the eluted peaks of the identified phytochemical compounds was tested statistically using a one-sample *t*-test and the results indicated no statistical difference (*p* > 0.05) with the injected hydrocarbon standards. However, when the percentage of the PGEO constituents was tested statically using ANOVA statistical test; the results indicated a statistical difference (*p* < 0.05) between the phytochemical constituents of the EO. 

### 2.2. Antioxidant Activity

In this study, the PGEO showed strong anti-DPPH radical activity (IC_50_ = 3.88 ± 0.45 µg/mL) and had 48.45% of the antioxidant potential of the commercial used antioxidant compound Trolox (IC_50_ = 1.88 ± 0.45 µg/mL), as presented in Table 2.

### 2.3. Target Metabolic Enzyme Inhibitory Activity

In a dose-dependent manner, the PGEO inhibited porcine pancreatic lipase, α-amylase, and α-glucosidase were compared with the antiobesity drug Orlistat and the antidiabetic pharmaceutical Acarbose, which were employed in the current study as positive controls. IC_50_ values are presented in Table 2. An independent sample *t*-test was performed for all biological activities compared with the positive control. The results indicated the positive control was more active with a significant difference (*p* < 0.05) for all the tests except for DPPH, which indicated no difference in the antioxidant activity compared with the Trolox positive control. 

### 2.4. Cytotoxicity 

The MTS assay was used to assess the cytotoxicity of five different concentrations of PGEO against HeLa, MCF-7, and Hep3B tumor cells. The cytotoxicity of five different concentrations of PGEO against HeLa, MCF-7, and Hep3B tumor cells was tested using the MTS assay. As indicated in Figure 2 and Table 2, the IC_50_ values for PGEO were 32.71 ± 1.25, 40.71 ± 1.89, and 315.19 ± 20.5 μg/mL against MCF-7, Hep3B, and HeLa, respectively.

### 2.5. Cyclooxygenase Inhibitory Activity

The COX (ovine/human) Inhibitor Screening Assay Kit (Cayman Chemical, MI, USA) was used to assess the COX-1/2 inhibitory activities of PGEO. Table 2 displays the IC_50_ values calculated from the experimental data. The results attained for the standard NSAIDs matched those described by the kit’s manufacturer. The tested PGEO inhibited COX1 isoenzymes better than COX2 iso-enzymes, with IC_50_ values of 14.03 and 275.97 µg/mL, respectively.

### 2.6. Antimicrobial Effect

The broth microdilution assay evaluated the antimicrobial effect of PGEO against seven microbial species. The results revealed that the EO had broad-spectrum antibacterial and antifungal effects against all the screened bacterial and fungal strains. However, PGEO had MIC values of 0.78 ± 0.01–100 ± 1.01 µg/mL, as demonstrated in Table 3.

## 3. Discussion

Recently, about 3000 types of natural EOs have been characterized globally, and only 10% of them are approved by the US FDA as safe products and are utilized in the pharmaceutical, cosmetics, and food industries. EOs have a diverse set of pharmacological properties, including anticancer, antidiabetic, insecticidal, antiparasitic, antimycotic, antiviral, antioxidant, and antimicrobial activity [19].

### 3.1. Phytochemistry of the Pelargonium graveolens EO

The PGEO contains approximately 120 phytochemicals, of which citronellol, geraniol, and linalool and their esters account for more than 60% of the total EO and are responsible for its odor [20]. PGEO is one of the costliest EOs used in perfumery, flavoring, and cosmetics. The hydrodistillation of air-dried *P. graveolens* leaves produced a colorless oil with a yield of 1.01% that was equal to the oil derived from *P. graveolens* aerial parts collected in Isfahan, Iran [21]. In contrast, Afifi et al. reported a 1.5% yield for the extraction of PGEO from its air-dried leaves collected in Amman, Jordan [22]. This variance can be due to the climate of Tulkarem, which lies 15 km east of the Mediterranean. Summers are long, hot, very humid, and arid in Tulkarem, while winters are cold and wet [23].

The PGEO was composed predominantly of citronellol (24.44%), citronellyl formate (15.63%), γ-eudesmol (8.60%), iso-menthone (7.43%), geranyl formate (3.40%), geranyl butanoate (2.69%), germacrene D (2.56%), phenylethyl tiglate (2.53%), linalool (2.45%), geranyl tiglate (2.38%), viridflorene (2.20%), and *cis*-rose oxide (2.04%). Geraniol, rhodinol, β-caryophellene, geranyl propanoate, citronellyl butanoate, γ-cadinene, and citronellyl tiglate were also present in noteworthy amounts. Other components of the tested oil were found in slight amounts (<1%). The prevailing group of oil constituents was oxygenated monoterpenes, accounting for 75.51% of total EO (Table 1), while monoterpene hydrocarbons existed with six compounds accounting for only 0.39% of the oil yield. Sesquiterpene hydrocarbons and oxygenated sesquiterpenes were present in 13.04% and 10.53%, respectively. These outcomes are consistent with previous findings by Ghannadi et al., who reported that citronellol (36.4%) and citronellyl formate (12.1%) were major constituents of the EO extracted from aerial parts of *P. graveolens* gathered in Isfahan Province and Central Iran [21]. Furthermore, Džamić et al. [24] reported the identification of 55 phytochemicals from PGEO, the major constituents of which are citronellol (24.54%), geraniol (15.33%), citronellyl formate (10.66%), and linalool (9.80%). Overall, all studies found that oxygenated monoterpenes were the most prevalent EO components, accounting for more than 60% of total EO yield. In contrast, monoterpene hydrocarbons existed with few composites, accounting for less than 2% of the total EO yield. All previous studies agree that citronellol is the most abundant compound in PGEO, but the order of the following five compounds varies [20,21,24,25,26,27]. This variation can be attributed to climate, humidity, soil, seasonal variation, and plant physiological conditions, as well as the EO distillation method [20].

### 3.2. Antioxidant Activity

An in vitro method that mimics the oxidation-reduction reactions that occur in living organisms was used to evaluate the EO’s antioxidant capacity. In terms of antioxidant activity, PGEO was able to reduce DPPH radicals to DPPH-H form in a dose-dependent manner. The EO displayed remarkable antioxidant properties, with an IC_50_ dose of 3.88 ± 0.45 µg/mL and 48.45% antioxidant activity compared with Trolox (IC_50_ = 1.88 ± 0.45 µg/mL). The antioxidant capabilities of the PGEO employed in this investigation were higher than those of PGEO from other countries [24,26,28].

### 3.3. Metabolic Lipase, α-Amylase, and α-Glucosidase Inhibitory Activities

Overweight, obesity, diabetes, and dyslipidemia are among worldwide health issues that are becoming more prevalent [29]. Obesity and diabetes have been linked to an increase in cancer mortality as well as cardiovascular, pulmonary, hepatic, and renal disease death rates. The significant frequency of these metabolic illnesses is clear and concerning in Palestine and many other nations in the last three decades. In reality, a pharmaceutical medication that inhibits the activity of metabolic enzymes such as α-amylase, α-glucosidase, and lipase can almost certainly heal these life-threatening disorders [30]. Aromatic plants have traditionally been utilized in various ethnopharmacological global systems of medicine to treat obesity, overweightness, dyslipidemia, and diabetes [31].

PGEO exhibited a modest lipase inhibitory effect at concentrations of 50, 100, 200, 300, and 400 μg/mL, with inhibition occurring in a dose-dependent manner and an IC_50_ dose of 478.14 ± 1.2 μg/mL; however, the positive control (Orlistat) had an anti-lipase with an IC_50_ dose of 12.3 ± 0.33 μg/mL. 

The *P. graveolens* plant, among other ethnomedicinal herbs, has long been used in folk medicine to treat diabetes. As a result, the current research looked into the ability of PGEO to inhibit inhibitors of carbohydrate metabolic enzymes such as α-amylase and α-glucosidase. The results showed that the tested EO suppressed α-amylase nearly 10-fold less than the positive control, Acarbose, with IC_50_ values of 66.09 ± 1.43 and 6.64 ± 1.32 µg/mL, respectively, at the tested concentrations (10, 50, 70, 100, and 500 µg/mL).

Notably, PGEO inhibited α-glucosidase in a concentration-dependent manner at concentrations (100, 200, 300, 400, and 500 mg/mL), with an IC_50_ values of 52.44 ± 0.29 compared with the positive control, Acarbose (IC_50_ = 37.15 ± 0.33 μg/mL).

Only one study was found that reported the PGEO α-amylase, α-glucosidase, and lipase inhibitory effects from Jordan. Afifi et al. [22] found that the PGEO also had a little anti-lipase effect compared with Orlistat, with IC_50_ values of 207.4 ± 5.2 and 0.114 ± 0.0 µg/mL, respectively, as well as another study from Iraq, showed that the IC_50_ of PGEO against the α-glucosidase enzyme, was 93.72 ± 4.76 μg/mL [15]. The current study is the first to look into the α-amylase and α-glucosidase inhibitory properties of PGEO.

### 3.4. Antimicrobial Effects

Aromatic plants and their EOs have several pharmacological characteristics that can be utilized to treat and prevent various human systemic illnesses, including communicable diseases. Many studies have reported that EOs can be used as supportive therapies for infectious diseases, cancer, diabetes, hypercholesteremia, and hyperlipidemia. Global healthcare systems have recently suffered from antimicrobial resistance (AMR) problems, especially in hospitals, which can affect anyone in any region of any age. AMR has the potential to jeopardize the active curing and prevention of a variety of infectious diseases caused by fungi, bacteria, viruses, and parasites. 

According to WHO statistics, antimicrobial resistance (AMR) is one of the primary causes of treatment regimen failure in infectious diseases. Secondary plant metabolites, such as EOs, can help to reduce or eliminate this problem.

Compared with the antimicrobial drugs Ampicillin, Ciprofloxacin, and Fluconazole, the antimicrobial test results revealed that PGEO had a broad-spectrum antimicrobial effect and repressed the growth of all the tested microbial strains. PGEO inhibited MRSA growth more effectively than Ampicillin and Ciprofloxacin, with MIC values of 1.56 ± 0.01, 32 ± 0.64, and 12.5 ± 0.79 µg/mL, respectively. The tested EO reduced *S. aureus* growth even more than Ampicillin and had the same inhibitory efficacy as Ciprofloxacin at MIC doses of 0.78 ± 0.01, 6.25 ± 0.2, and 0.78 ± 0.01 µg/mL, respectively. Furthermore, PGEO displayed strong anticandidal activity compared with Fluconazole, with MIC values of 6.25 ± 0.02 and 3.12 ± 0.01 µg/mL, respectively.

MRSA is a deadly nosocomial infection found in most countries, with about 10–20% of all hospital and healthcare patients suffering from this highly infectious disease [32]. *S. aureus* is a Gram-positive bacterium linked to food poisoning as a result of the consumption of food containing enterotoxins, and several nations have reported a high number of food poisoning cases caused by *S. aureus*. This bacteria is also responsible for a wide range of diseases, including sepsis, bacteremia, toxic shock syndrome, endocarditis, osteomyelitis, meningitis, and pneumonia, as well as superficial skin infections such as abscesses, scalded skin syndrome, pimples, carbuncles, folliculitis, cellulitis, impetigo, and boils [33].

As noticed, PGEO from Palestine exhibited better antimicrobial activities than PGEOs from various countries [25,27,34]. According to Lopez-Romero et al., citronellol showed potent bactericidal properties against *S. aureus* and *E. coli* pathogens, with MIC values of 0.40 and 0.30 mg/mL [35]. Guimares et al. discovered that citronellol had strong bactericidal activity against *S. aureus* species, with an MIC value of 0.03 mg/mL [36].

### 3.5. Cytotoxicity

The current study used cultured HeLa, MCF-7, and Hep3B tumor cell lines to assess the effect of PGEO on human cancer cells using the trypan blue MTS assay. The results showed that incubating tumor cells for 24 h with PGEO (500, 120, 60, 30, and 10 µg/mL) reduced their viability in a dose-dependent manner.

The number of dead cells increased as the concentration of PGEO increased. PGEO had the highest cytotoxic effect at 500 µg/mL, with percentages of 56.57, 98.48, and 94.41% against HeLa, MCF-7, and Hep3B tumor cell lines, respectively. The utmost cytotoxicity was observed against MCF-7, followed by Hep3B and HeLa cancer cells, with IC_50_ doses of 32.29 ± 1.01, 44.60 ± 1.71, and 324.27 ± 2.2 µg/mL, respectively, compared with doxorubicin anticancer drug (positive control), which had cytotoxic IC_50_ doses of 0.37 ± 0.22, 1.21 ± 1.0, and 0.84 ± 1.1 µg/mL, respectively.

Fayed [26] discovered that the PGEO from Egypt displayed cytotoxic properties against NB4 (acute promyelocytic leukemia) and HL-60 (acute myeloid leukemia), with IC_50_ values of 62.50 and 86.5 µg/mL, respectively. Halees et al. [37] reported that *P. graveolens* aqueous extract from Jordan had cytotoxic activity against breast cancer EMT6/P, MCF-7, and T47D cells, with IC_50_ doses of 4105, 590, and 610 µg/mL, respectively. As a result, we can conclude that PGEO from Palestine has a more potent cytotoxic effect on breast cancer MCF-7 cells than *P. graveolens* aqueous extract from Jordan, which has IC_50_ values of 32.29 and 590 µg/mL, respectively. In another recent study, PGEO showed the greatest antiproliferative activity on the AGS cell lines with an inhibition rate of 92.87 ± 0.13% at the highest dose (4 μL/mL), followed by the MV3 line (88, 76 ± 0.96%) [38].

Yu et al. investigated the mechanism of action of citronellol and concluded that the major molecules of the PGEO can induce necroptosis in NCI-H1299 cells by upregulating TNF-expression and RIP3 and RIP1 activities, downregulating caspase-8 and caspase-3 activities and inducing overactivated PARP. The study also demonstrated that after citronellol treatment, there were two ROS accumulation pathways: one directed the stimulation of NCI-H1299 cells and the other directed the necroptosis process. Citronellol is expected to be used to treat tumors in the future [39].

### 3.6. Cyclooxygenase Suppressant Effect

COX has two distinct isomers, COX-1 and COX-2, that participate in converting arachidonic acid to prostaglandin. COX-1 can protect the mucosal membranes of the gastrointestinal tract from ulcers, preserve blood flow in kidneys that are failing, and limit platelet oxygenation, whereas COX-2 is generally missing from cells but is generated in reaction to inflammation [40].

The current study was designed to evaluate the effect of PGEO on the inhibitory activity of the COX enzyme, as traditional NSAIDs are not specific in inhibiting both COX isoforms, and no previous studies have investigated the effect of PGEO on the COX inhibitory activity until now. On the other hand, PGEO was found to have anti-inflammatory or COX-inhibitory properties. The tested PGEO exhibited selective inhibition activity toward the COX-1 enzyme, with an IC_50_ value of 14.034 µg/mL. In addition, it showed weak inhibitory activity toward the COX-2 enzyme, with an IC_50_ value of 275.97 µg/mL. However, the COX-1 and COX-2 IC_50_ values of the positive control celecoxib were 5.72 and 0.0152 µg/mL, respectively. The PGEO, with a low COX-1 IC_50_ value and a high COX-2 IC_50_ value, is considered a promising analgesic and anti-inflammatory agent with low platelet abnormalities, renal dysfunction, and gastrointestinal damage.

## 4. Materials and Methods

### 4.1. Plant Material

In June 2020, the fresh *P. graveolens* leaves were collected in Tulkarem, Palestine. Dr. Nidal Jaradat authenticated the plant, which was then deposited in the Herbal Product Laboratory at An-Najah National University, with the dried sample kept under the voucher specimen number (Pharm-PCT-2779). The leaves were rinsed in distilled water many times before being dried in the shade at room temperature and normal level of humidity. The powdered dried leaves were then kept in amber-colored glass jars.

### 4.2. Essential Oil Extraction

The PGEO was extracted by the water-distillation system according to the standard method described in the European Pharmacopoeia [41,42]. In brief, about 100 g of the herb was mixed well with 1 L of distilled water, and the PGEO was extracted using a Clevenger apparatus (Merck, Kenilworth, NJ, USA) operating at atmospheric pressure for 180 min at 100 °C. The extracted VO was chemically dried using MgSO_4_ and the obtained EO yield from the dried plant sample was 1.01%. The PGEO was stored in a refrigerator at 4 °C in a dark chamber until further use. 

### 4.3. Chromatographic Analyses 

The EO was analyzed using an HP 5890 series II gas chromatograph (Hewlett Packard, Palo Alto, CA, USA) equipped with a fused-silica capillary column (0.25 mm × 30 m, the film thickness of 0.25 µm) and linked to a Perkin Elmer Elite-5-MS operating with an ionizing voltage of 70 eV (Perkin Elmer, Waltham, MA, USA). At a flow rate of 1.1 mL/min, helium served as the carrier gas. The injector temperature was set to 250 °C, the oven temperature was held at 50 °C for 5 min before ramping up to 280 °C at a rate of 4.0 °C/min, and the MS detector was set to 250 °C. The solvent delay was 0–4.0 min, while the MS scan time was 4–62.5 min, spanning a mass range of 50.00–300.00 *m*/*z*. The EO contents were identified by comparing retention indices (RIs) calculated using a standard mixture of *n*-alkanes (C_6_–C_27_) (Sigma-Aldrich, Schnelldorf, Germany) and mass spectra fragmentation patterns to those obtained from authentic samples and/or the NIST (National Institute of Standards and Technology, NIST Version 2.0, Gaithersburg, MD, USA)

### 4.4. Free Radical Scavenging Activity

A mixture of PGEO and methanol (1 mg/mL) was serially diluted with methanol to generate different concentrations (2, 5, 10, 20, 50, and 100 µg/mL). A total of 1 mL of EO solution was combined with 2 mL of a methanolic solution of 2,2-diphenyl-1-1-picrylhydrazyl (DPPH, Sigma-Aldrich, Schnelldorf, Germany) and incubated for 30 min at room temperature under light exclusion. Trolox (Sigma-Aldrich, Burlington, MA, USA) was utilized as a positive control, and the blank solution was generated by substituting the EO solution with methanol. Spectrophotometry at 517 nm was used to measure the absorbance, which was then compared with the control. The DPPH suppressant effect was calculated using the formula below:$$I\;\left(\%\right)=\frac{\mathrm{ABSblank}-\mathrm{ABStest}}{\mathrm{ABSblank}}\times 100$$
where I (%) is the percentage of antioxidant activity [43,44,45].

### 4.5. Porcine Pancreatic Lipase Inhibitory Activity

The anti-lipase effect of PGEO was conducted according to the method of [46,47]. The EO was serially diluted to 50, 100, 200, 300, and 400 µg/mL concentrations. Furthermore, a 1 mg/mL stock solution of pancreatic lipase (Sigma, St. Louis, MO, USA) in 10% DMSO and a solution of p-nitrophenyl butyrate (PNPB) (Sigma-Aldrich, Schnelldorf, Germany) and acetonitrile (10.49 mg/mL) were prepared. The absorbance was measured with a UV-Vis spectrophotometer (Shimadzu-UV-1800, Nakagyo-ku, Japan) at 405 nm, and the anti-lipase effect was calculated using the following equation: $$\mathrm{lipase}\;\mathrm{enzyme}\;\mathrm{inhibitory}\%\;=\frac{\mathrm{ABSblank}-\mathrm{ABStest}}{\mathrm{ABSblank}}\times 100$$

### 4.6. α-Amylase Inhibitory Activity

The PGEO α-amylase inhibitory activity was estimated according to the method reported by Ali et al. [48] with minor modifications. The EO was dissolved in DMSO (Riedel-de-Haen, Hamburg, Germany) and then diluted with a buffer (Na_2_HPO_4_/NaH_2_PO_4_ (0.02 M), NaCl (0.006 M) at pH 6.9) to a concentration of 1000 µg/mL. Concentration series of 10, 50, 70, 100, and 500 µg/mL were prepared. A total of 0.2 mL of porcine pancreatic α-amylase enzyme solution (Sigma-Aldrich, St. Louis, MO, USA) with a concentration of 2 units/mL was mixed with 0.2 mL of the PGEO and incubated at 30 °C for 10 min. After that, 0.2 mL of freshly prepared starch solution (1%) was added and the mixture was incubated for at least 3 min. The reaction was stopped by the addition of 0.2 mL dinitrosalicylic acid (DNSA) (Alfa-Aesar, Lancashire, UK). The mixture was then diluted with 5 mL of distilled water and heated in a water bath at 90 °C for 10 min. The mixture was left to cool down to room temperature, and the absorbance was measured at 540 nm. A blank was prepared following the same procedure by replacing the PGEO with 0.2 mL of the buffer. Acarbose (Sigma-Aldrich, Burlington, MA, USA) was utilized as a positive control and prepared to adopt the same procedure described above. The α-amylase inhibitory activity was calculated using the following equation: $$\alpha -\mathrm{Amylase}\;\%=\frac{\mathrm{ABSblank}-\mathrm{ABStest}}{\mathrm{ABSblank}}\times 100$$

### 4.7. α-Glucosidase Inhibitory Activity

Based on the usual technique [30], the α-glucosidase inhibitory activity of the PGEO was determined. In each test tube, a reaction mixture containing 20 μL of varying concentrations of PGEO (100, 200, 300, 400, and 500 µg/mL), 10 μL of α-glucosidase (1 U/mL), and 50 μL of phosphate buffer (100 mM, pH = 6. 8), at 37 °C was incubated for 15 min. Using a UV-Vis spectrophotometer, the absorbance at 405 nm of the released p-nitrophenol was determined. Acarbose was used as a positive control. The PGEO inhibition percentage was estimated utilizing the following equation:$$\alpha -\mathrm{Glucosidase}\;\%=\frac{Ab-As}{Ab}\times 100$$
where *A_b_* is the absorbance of the blank and *A_S_* is the absorbance of the tested sample or control.

### 4.8. Antimicrobial Activity

The microdilution method was used to assess the antimicrobial activity of the PGEO against selected bacterial and fungal strains according to the method [49]. American Type Culture Collection (ATCC) bacterial species, including *Pseudomonas aeruginosa* (9027), *Escherichia coli* (25,922), *Klebsiella pneumonia* (13,883), *Proteus vulgaris* (8427), *Staphylococcus aureus* (2592), and a diagnostically confirmed clinical strain methicillin-resistant *S. aureus* (MRSA) were utilized to assess the bacteriostatic effect. The antifungal effect of the PGEO was estimated against the growth of *Candida albicans* (ATCC 90028). 

### 4.9. Cell Culture and Cytotoxicity Assay

Human cervical (HeLa), breast (MCF-7), and liver (Hep3B) cancer cell lines were cultured in RPMI-1640 medium (Sigma-Norwich, Norfolk, UK). Then each kind of the tested cells was treated with 10% fetal bovine serum mixed with 1% penicillin/streptomycin antibiotics (BI, Ghaziabad, UP, India), and 1% L-glutamine (Sigma-Norwich, Norfolk, UK). The cancer cells were cultured at 37 °C in a humidified atmosphere with 5% CO_2_ and then planted in a 96-well plate at 2.6 × 10^4^ cells/well. After 2 days, the EO with different concentrations (10, 30, 60, 120, and 500 µg/mL) was applied to the cells for 24 h. According to the manufacturer’s guidelines, cell viability was estimated by the CellTilter 96^®^ Aqueous One Solution Cell Proliferation (MTS) method. In the end, 100 μL of media and 20 μL of MTS solution were added to each well, and the well plates were incubated at 37 °C for 2 h. The absorbance was measured at 490 nm [50,51].

### 4.10. Cyclooxygenase Inhibitory Effect 

The ability of the PGEO to prevent the conversion of arachidonic acid (AA) to Prostaglandin H2 by human recombinant COX-2 and bovine COX-1 was assessed using a COX inhibitor screening assay kit according to the manufacturer’s guidelines (Cayman Chemical, MI, USA). The IC_50_ of COX-1/COX-2 activity of the PGEO was determined with the assay run, in duplicate, with various concentrations. A standard curve of eight concentrations of prostaglandin, a non-specific binding sample, and a maximum binding sample was used, as instructed in the kit manual, to determine the inhibition of the sample plant, applying the generated multiple regression best-fit line. The inhibition percent of the used concentration was utilized to estimate the IC_50_ [52].

### 4.11. Data Analysis

The antioxidant, antiobesity, antidiabetic, cytotoxic, and COX inhibitory tests of the PGEO were performed in triplicate. The results were expressed as mean ± the standard deviation (SD). Statistical Package for the Social Sciences (SPSS) software program (SPSS Inc., Chicago, IL, USA) was used to perform statistical analysis. One and two sample *t*-tests were used to test for significant difference for one and two means, respectively. An analysis of variance (ANOVA) test was used to test if there was a statistically significant difference between means. In all the mentioned statistical tests, if the *p*-value is less than 0.05, the null hypothesis was rejected and it was concluded that a significant difference does exist.

## 5. Conclusions

For the first time, the EO of PG leaves from Palestine was studied qualitatively and quantitatively. The results revealed that the tested essential oil displayed potent antioxidant, anti-glucosidase, and antimicrobial properties, particularly against MRSA, *S. aureus*, and *C. albicans*. Furthermore, it demonstrated potent cytotoxicity against MCF-7, Hep3B, and HeLa cancer cell lines, as well as selective inhibition activity against the COX-1 enzyme. These findings suggest that *PGEO* may be a valuable source of natural antioxidants that can aid in the detoxification mechanisms of living organisms, particularly in the prevention of oxidative stress-related diseases. Furthermore, it can be used as a supportive and alternative therapy for infectious diseases caused by harmful pathogens. Moreover, it has intriguing cytotoxic and COX-1 inhibitory properties. In this case, the use of PGEO as a new therapeutic agent with functional properties for food, food supplements, and pharmaceuticals should be investigated.

## Figures and Tables

**Figure 1 molecules-27-05721-f001:**
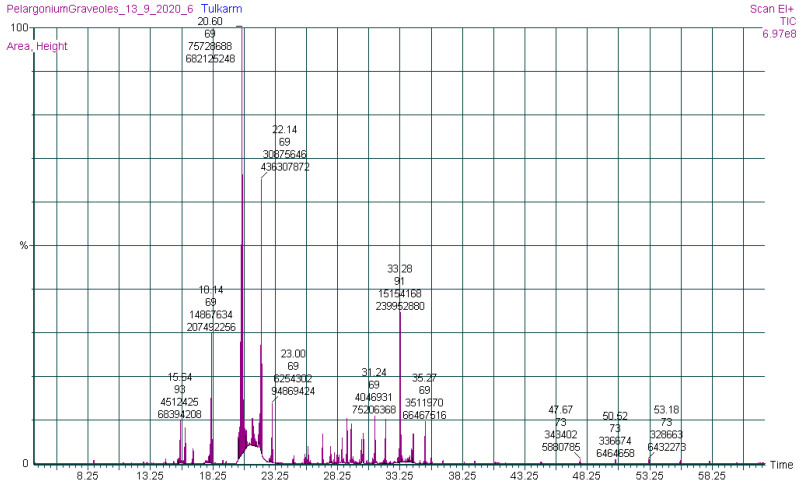
GC-MS chromatogram of PGEO.

**Figure 2 molecules-27-05721-f002:**
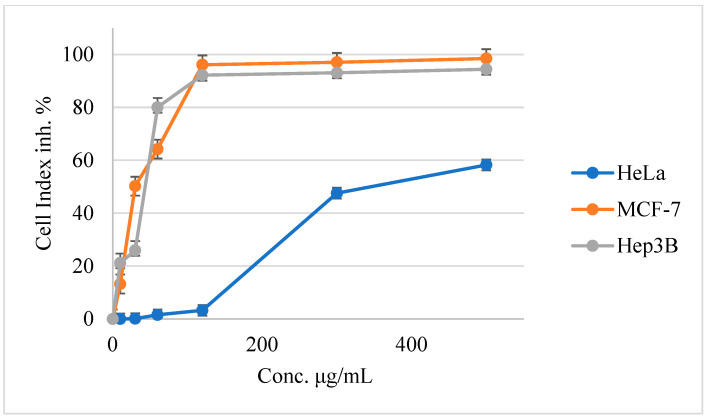
Cytotoxicity of PGEO was determined using an MTS assay against HeLa, MCF7, and Hep3B cancer cells.

**Table 1 molecules-27-05721-t001:** The chemical components of PGEO.

Constituents	Retention Times	Retention Indices	Area	EO (%)
*α*-Pinene	8.71	933	5,219,705	0.19
Myrcene	11.09	991	1,247,632	0.04
*α*-Phellandrene	11.73	1006	809,821	0.03
*p*-Cymene	12.5	1018	599,766	0.02
Limonene	12.57	1022	244,234	0.01
Sylvestrene	12.69	1030	2,322,006	0.08
Santolina alcohol	12.91	1035	761,560	0.03
Phenylacetaldehyde	13.16	1041	142,181	0.01
*o*-Tolualdehyde	13.34	1046	1,297,560	0.05
trans-Ocimene	13.45	1048	684,117	0.02
*cis*-1,1,3,5-tetramethylcyclohexane	13.945	1060	225,932	0.01
*cis*-Linalool oxide	14.47	1073	8,052,599	0.29
trans-Linalool oxide	15.45	1097	7,288,440	0.26
Linalool	15.65	1102	68,394,208	2.45
*cis*-Rose oxide	16.033	1111	56,869,124	2.04
trans-Rose oxide	16.67	1128	22,926,890	0.82
Neoisopulegol	17.52	1151	1,362,589	0.05
Citronellal	17.67	1155	3,737,836	0.13
trans-Menthan-3-one	17.79	1158	3,701,619	0.13
Isomenthone	18.14	1167	207,492,256	7.43
Neoisomenthol	19.05	1191	4,323,271	0.15
α-Terpineol	19.31	1197	4,471,245	0.16
Rhodinol	20.31	1224	43,395,516	1.55
Citronellol	20.6	1233	682,125,248	24.44
Neral	20.92	1242	2,910,552	0.1
Geraniol	21.42	1256	43,679,076	1.57
Citronellyl formate	22.14	1276	436,307,872	15.63
Geranyl formate	23	1300	94,869,424	3.4
*α*-Cubebene	24.65	1346	5,753,072	0.21
Citronellyl acetate	24.74	1348	12,922,121	0.46
*α*-Ylangene	25.61	1373	15,292,231	0.55
Geranyl acetate	25.69	1375	9,889,024	0.35
Bourbonene	25.88	1379	26,907,980	0.96
*β*-Elemene	26.07	1385	4,791,249	0.17
Longifolene	26.66	1410	1,218,996	0.04
*β*-Caryophellene	27.05	1421	46,656,692	1.67
*β*-Copaene	27.38	1433	1,369,444	0.05
Citronellyl propanoate	27.67	1442	27,053,746	0.97
6,9-Guaiadiene	27.74	1444	13,679,959	0.49
Spirolepechinene	27.83	1447	923,947	0.03
trans-Muurola-3,5-diene	27.98	1451	17,618,874	0.63
trans-Prenyl limonene	28.19	1458	13,135,552	0.47
Aromadendrane	28.34	1463	13,325,749	0.48
Geranyl propanoate	28.6	1464	39,640,280	1.42
*γ*-Muurolene	28.72	1472	7,682,250	0.28
Amorphene	28.82	1475	1,448,632	0.05
Germacerene D	29	1484	71,366,280	2.56
*β*-Selinene	29.1	1488	1,581,278	0.06
Viridiflorene	29.33	1495	61,363,196	2.2
Bicyclogermacren	29.47	1499	10,318,428	0.37
*α*-Muurolene	29.554	1502	3,793,223	0.14
trans-β-Guaiene	29.689	1507	2,964,878	0.11
Germacrene A	29.794	1910	2,806,741	0.1
*γ*-Cadinene	29.994	1517	6,085,463	0.22
*Δ*-Cadinene	30.159	1522	36,284,956	1.3
Citronellyl butanoate	30.314	1528	46,489,824	1.67
*α*-Cadinene	30.584	1537	4,487,362	0.16
*α*-Agarofuran	31.054	1553	8,448,215	0.3
Geranyl butanoate	31.239	1559	75,206,368	2.69
11-Norbourbonan-1-one	31.384	1563	2,648,644	0.09
*β*-Phenyl ethyl tiglate	32.09	1588	70,742,752	2.53
Geranyl 2-methyl butanoate	32.455	1600	3,184,698	0.11
*β*-Atlantol	32.76	1611	2,970,958	0.11
1,10-di-epi-Cubenol	33.06	1621	7,473,155	0.27
*γ*-Eudesmol	33.275	1629	239,952,880	8.6
Amorph-4-en-7-ol	33.451	1636	10,539,318	0.38
7-epi-a-Eudesmol	34.175	1661	21,753,554	0.78
Citronellyl tiglate	34.311	1667	46,228,004	1.66
Geranyl tiglate	35.261	1700	66,467,516	2.38
E-Nerolidyl acetate	35.721	1716	8,573,079	0.31
Total				99.47
**Phytochemical group**				
Monoterpene hydrocarbon				0.39
Oxygenated monoterpenoid				75.51
Sesquiterpene hydrocarbons				13.04
Oxygenated sesquiterpenes				10.53
Total				99.47

**Table 2 molecules-27-05721-t002:** The IC_50_ (µg/mL) for PGEO against DPPH, lipase, α-amylase, α-glucosidase, COX-1, and COX-2, as well for the positive controls.

Antioxidants, Target Metabolic Enzymes, Cancer Cells Lines, and COX	IC_50_ (µg/mL)	
	PGEO	Positive Controls
**DPPH**	3.88 ± 0.45	1.88 ± 0.45 ^a^
**Lipase**	478.14 ± 1.2	12.3 ± 0.33 ^b^
**α-Amylase**	66.09 ± 1.43	6.64 ± 0.32 ^c^
**α-Glucosidase**	52.44 ± 0.29	37.15 ± 0.33 ^c^
**HeLa**	315.19 ± 2.05	0.84 ± 1.1 ^d^
**MCF7**	32.71 ± 1.25	0.37 ± 0.22 ^d^
**Hep3B**	40.70 ± 1.89	1.21 ± 1.0 ^d^
**COX-1**	14.03 ± 1.74	5.72 ± 0.09 ^e^
**COX-2**	275.97 ± 2.08	0.0152 ± 0.007 ^e^

^a^ Trolox, ^b^ Orlistat, ^c^ Acarbose, ^d^ Doxorubicin, and ^e^ Celecoxib.

**Table 3 molecules-27-05721-t003:** Antimicrobial effects (MICs) of PGEO, Ciprofloxacin, Ampicillin, and Fluconazole (µg/mL).

Microbial Species	PGEO	Ampicillin	Ciprofloxacin	Fluconazole
** *S. aureus* **	0.78 ± 0.01	6.25 ± 0.2	0.78 ± 0.01	-
**MRSA**	1.56 ± 0.01	32 ± 0.64	12.5 ± 0.79	-
** *E. coli* **	50 ± 0.56	3.12 ± 0.56	0.78 ± 0.01	-
** *P. vulgaris* **	6.25 ± 0.12	3.25 ± 0.21	0.06 ± 0.001	-
** *K. pneumoniae* **	25 ± 0.31	12.5 ± 0.05	0.06 ± 0.001	-
** *P. aeruginosa* **	50 ± 0.55	100 ± 1.01	3.12 ± 0.31	-
** *C. albicans* **	6.25 ± 0.02	-	-	3.12 ± 0.01

## Data Availability

All data are contained within the article.

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
