# Peer review of "Chemical Markers and Pharmacological Characters of Pelargonium graveolens Essential Oil from Palestine"

_molecules, 2022, doi:10.3390/molecules27175721_

Round 1

Reviewer 1 Report

In this study, the essential oil from leaves of Pelargonium graveolens from Palestine was tested for biological activities (cytotoxic, antioxidant, antimicrobial, metabolic, and inhibition of cyclooxygenase (COX) enzymes). Given that this oil has been examined a lot before, the only novelty in the research is the testing of alpha amylase, alpha glucosidase and COX inhibitory activities. However, some of the obtained results are very good, so they would be significant for the scientific community. Therefore, I suggest that the manuscript should be accepted with minor modifications. 

The title of the manuscript and the abstract are too long. I suggest that they be shortened. This especially applies to the abstract.

The same reference was cited in Lines 227-229 and Lines 240-242. One of these sentences should be deleted.

Lines 327-331 This part is unnecessary in the text. Delete it

In the Material and methods, check whether the method for determining alpha amylase inhibitory activity is well written.

Author Response

Dear Editor,

Many thanks for providing us with the opportunity to revise our manuscript. We would also like to thank you and the reviewers for the time and expertise in providing feedback.

We think that all comments raised by the reviewers are legitimate and requires consideration. We would like to profoundly thank them for their constructive comments which have greatly improved the manuscript.   

Please find below our response to reviewers comments. We have considered carefully all of the comments and have amended the manuscript as appropriate. The amended text is highlighted in red font throughout the manuscript. We have provided a detailed response to each of the comments.

Reviewer 1 comments

In this study, the essential oil from leaves of Pelargonium graveolens from Palestine was tested for biological activities (cytotoxic, antioxidant, antimicrobial, metabolic, and inhibition of cyclooxygenase (COX) enzymes). Given that this oil has been examined a lot before, the only novelty in the research is the testing of alpha amylase, alpha glucosidase and COX inhibitory activities. However, some of the obtained results are very good, so they would be significant for the scientific community. Therefore, I suggest that the manuscript should be accepted with minor modifications. 

The title of the manuscript and the abstract are too long. I suggest that they be shortened. This especially applies to the abstract.

Done for title and abstract

The same reference was cited in Lines 227-229 and Lines 240-242. One of these sentences should be deleted.

Done

Lines 327-331 This part is unnecessary in the text. Delete it

Done

In the Material and methods, check whether the method for determining alpha amylase inhibitory activity is well written.

We rewrote this assay to be more clear for readers.

Reviewer 2 Report

Dear authors, 

I have completed my review of the submission by Jaradat et al. It is a potential submission towards Molecules. The results are novel and interesting and worth being published in Molecules with its associated Impact Factor and Readership. However, I see some revisions are needed to improve the manuscript:

General comments:

A- Text must be revised carefully for typos and language mistakes,

B- Results Introduction and Discussion sections have to be better fortified with recent published literature even in other species.

Specific comments: 

1-The title must be rephrased to be concise and informative,

2- Likewise, the abstract should be shortened but the following items must be stated briefly: background and objectives, methodology, main results and conclusions,

3-Line 21, please use the abbreviation PGEO throughout the entire paper after defining it at the first mention of course,

4- Results must be analyzed for statistical differences. For instance, Tables 2 and 3 as well as Fig.2. The outcomes are meaningless without statistical analyses. Even, I suggest the use of multivariate approaches such as PCA, correlations, regressions, etc to better exploit your results,   

5- I see that introduction and discussion sections have to be deepened to reflect the importance of the study. Here are some recently published to be added and discussed:10.1016/j.bcab.2022.102327; 10.33263/BRIAC112.93589371; 10.33263/BRIAC126.84418452),

 6- Line 269, please remove the repeated expression,

7- Line 340, and the entire manuscript, please use essential oil isolation instead of extraction, 

8- For plant material sampling,  please add a brief description of local pedoclimatic conditions due to their importance in explaining essential oil yield, profiling and related activities, 

9- For essential oil isolation, please add the required conditions , e.g time etc. A detailed description must be added but concisely. 

10-Line 433, please use mean instead of means: mean ± standard deviation,

11- For references section, please DOI.

Recommendation: accept after major revisions. 

Kind regards.

Author Response

Dear Editor,

Many thanks for providing us with the opportunity to revise our manuscript. We would also like to thank you and the reviewers for the time and expertise in providing feedback.

We think that all comments raised by the reviewers are legitimate and requires consideration. We would like to profoundly thank them for their constructive comments which have greatly improved the manuscript.   

Please find below our response to reviewers comments. We have considered carefully all of the comments and have amended the manuscript as appropriate. The amended text is highlighted in red font throughout the manuscript. We have provided a detailed response to each of the comments.

Reviewer 2 comments

I have completed my review of the submission by Jaradat et al. It is a potential submission towards Molecules. The results are novel and interesting and worth being published in Molecules with its associated Impact Factor and Readership. However, I see some revisions are needed to improve the manuscript:

General comments:

A- Text must be revised carefully for typos and language mistakes,

We polished the whole manuscript by native speaker Professor Nawaf Al-Maharek

B- Results Introduction and Discussion sections have to be better fortified with recent published literature even in other species.

We added a recent published papers to the mentioned sections that belongs to the last years 2021 and 2022. 

Specific comments: 

1-The title must be rephrased to be concise and informative,

We rephrased and concise the title

2- Likewise, the abstract should be shortened but the following items must be stated briefly: background and objectives, methodology, main results and conclusions,

We now shortened the abstract part as much as possible and we followed the Molecule journal style as the abstract section is not split into subsections.

3-Line 21, please use the abbreviation PGEO throughout the entire paper after defining it at the first mention of course,

Done

4- Results must be analyzed for statistical differences. For instance, Tables 2 and 3 as well as Fig.2. The outcomes are meaningless without statistical analyses. Even, I suggest the use of multivariate approaches such as PCA, correlations, regressions, etc to better exploit your results,   

Statistical testing for significant difference was performed for the above-mentioned results. SPSS statistical package program was used in this statistical analysis. Different types of analyses were used when appropriate; we used one and to two t-sample tests. We also used analysis of variance for comparing multi-means. The methodology section and the result section were revised accordingly.

5- I see that introduction and discussion sections have to be deepened to reflect the importance of the study. Here are some recently published to be added and discussed:10.1016/j.bcab.2022.102327; 10.33263/BRIAC112.93589371; 10.33263/BRIAC126.84418452),

We added the related references accordingly

 6- Line 269, please remove the repeated expression,

Done

7- Line 340, and the entire manuscript, please use essential oil isolation instead of extraction, 

Done

8- For plant material sampling,  please add a brief description of local pedoclimatic conditions due to their importance in explaining essential oil yield, profiling and related activities, 

Done

9- For essential oil isolation, please add the required conditions , e.g time etc. A detailed description must be added but concisely. 

Done

10-Line 433, please use mean instead of means: mean ± standard deviation,

Done

11- For references section, please DOI.

We used the Journal reference style, thank you for your comment.

Round 2

Reviewer 2 Report

Dear authors, 

I have gone through the revised version. It seems that the submission has been improved and therefore I recommend its publication within Molecules. 

Regards.